# EXPLICIT FOUNDATION MODEL OPTIMIZATION WITH SELF-ATTENTIVE FEED-FORWARD NEURAL UNITS

## ABSTRACT

Iterative differential approximation methods that rely upon backpropagation have enabled the optimization of neural networks; however, at present, they remain computationally expensive, especially when training models at scale. In this paper, we present a computationally efficient alternative for optimizing neural networks that can both reduce the costs of scaling neural networks and provide high-efficiency optimizations for low-resource applications. This paper will discuss how we derive a general result about feed-forward neural networks and then extend this solution to compositional (mult-layer) networks, which we then apply to a simplified transformer block, containing both feed-forward and self-attention layers. These developments lead us to train highly-specified and complex multi-layer neural architectures that we refer to descriptively as self-attentive feed-forward unit (SAFFU) layers, which we apply to our development of a hyper-efficient transformer, which appears to generalize well over small—cognitively-feasible—volumes of data. Results from testing demonstrate explicit solutions grossly outperform models optimized by backpropagation alone. Moreover, further application of backpropagation after explicit solutions leads to the discovery of better optima from smaller scales of data, i.e., that training highly-performant models from much smaller scales of data is enabled by warm starting models with their explicit solutions. Using the efficiency and consistency of the SAFFU's explicit solution, we carry out ablation experiments training a roadmap of about 250 transformer models over 1-million tokens, each, to determine ideal hyperparameterizations for the SAFFU-based transformer. We find that multiple different architectural variants of the SAFFU-transformer are capable of highly-performant models. Most critically, we discover from this ablation that some of the most performant models are in fact not the most parameterized. These results appear to strongly indicate that well-generalized models could be reached more efficiently (using less data) by using explicit solutions, and moreover, that architectural exploration using explicit solutions can pay dividends in guiding the search for efficient architectures containing fewer parameters, and which could be incorporated into low-resource hardware where AI might be embodied.

## 1 INTRODUCTION AND RELATED WORK

The cost of training large language models (LLMs) becomes extremely expensive when models become large in part due to large parameter requirements, but perhaps most of all from the tremendous scales of data required—LMs commonly require volumes of language that far exceed what a human would experience in a lifetime. Naturally, two concerns confront us: 1) training LLMs more efficiently, with respect to training times and computational costs; and 2) obtaining LLM-like abilities from smaller quantities of data, i.e., from at most what a human might experience. We show how *explicit solutions* to parameter optimization—which utilize assumptions over architectures to mathematically deduce algebraic forms for the parameters in neural network weight matrices—without backpropagation—make significant headway in satisfying concerns 1 & 2. Once an explicit solution is mathematically derived for a neural network, "plug and chug" computations can be leveraged to great efficiency to produce more-performant and -generalized models, using very little data.

Alongside escalating size and complexity, LLMs are becoming ever more *central* to applied work in artificial intelligence (AI). Superlative self-attention-based models in natural language processing

(NLP) now demonstrate capabilities attracting research interest and investment alongside counterparts in computer vision, like the diffusion probabilistic models (Ho et al., 2020) in DAll-E (Ramesh et al., 2021) and Stable Diffusion (Rombach et al., 2022). The potential to further amplify capabilities by combining text, images, and other modalities to construct even more powerful models, as exemplified by the likes of KOSMOS-1 (Huang et al., 2023) and GPT-4 (OpenAI, 2023), suggests staggering advancements may be on the cusp of development.

Still, our collective understanding of the inner workings of these models is far from complete. Limited understanding in the internal mechanisms of models hinders our ability to fully exploit their capabilities, while simultaneously raising challenges (Bommasani et al., 2022). Reliability and safety is a primary concern: LLMs are prone to generating biased and unreliable text, and diffusion models produce distorted images that conflict with basic human perception. The unpredictable behaviors of neural models in novel contexts challenges their operational benefits to humans via their (in)abilities to avoid inadvertent harms (Kenton et al., 2021; Weidinger et al., 2021; Tamkin et al., 2021; Hendrycks et al., 2023). Efficiency is also a major concern (Shen et al., 2023)—backpropagation is ubiquituous in optimization, and still entails a high computational cost, particularly as models scale over larger amounts of data (Rumelhart et al., 1986a;b), escalating processing requirements.

We ask: "how can these challenges can be overcome to ensure models are reliable, interpretable, and efficient?", and posit that understanding the optimization processes underlying these models is crucial. Perhaps, grasping the intricacies of model optimization will allow for a more straightforward approach, requiring fewer iterations to achieve the same or better quality results? Furthermore, understanding how models optimize allows us to adjust specific parameters in the weight matrices, enabling models to perform in a desired manner. Here, we extend our knowledge of explicit solutions from single-layer feed-forward neural networks, to an architecture with compositionally-linked feed-forward and self-attention layers. Our work demonstrates an explicit optimization technique that significantly accelerates model training processes, reaching optima far beyond the reach of backpropagation, alone. So when this solution is applied to self-attention networks, it accelerates time-to-optimization *and* finds vastly better optima with better generalization qualities, offering a vital alternative to the current trends in neural network training.

Explicit solutions relate to recent work focused on finding that attention layers converge in direction to SVM solutions (Tarzanagh et al., 2023) and that transformers may rediscover standard estimation algorithms (Akyürek et al., 2023). Explicit solutions also connect to recent discoveries finding generalization in overparametrized networks occurs beyond the point of dataset memorization (Power et al., 2022). Likewise, this work is also connected to efforts aimed at improving the overall training efficiency of transformers, such as one attention type developed to reduce memory reads/writes between GPU high bandwidth memory and on-chip SRAM (Dao et al., 2022).

By conducting ablation experiments over a large number of LM architectural variants, we discover that "warming up" (warm-start) models with the explicit solution for self-attention leads to better generalization, more rapidly. This discovery is largely invariant to the scales of training data utilized, i.e., warm-starts lead to objectively better models on both large and small data sets. Furthermore, our findings indicate that iterative optimization with backpropagation *only* leads to generalized models *with* the explicit solution—models initialized randomly at least appear to require more computation than any conducted experiments, regardless of scale. We conjecture that *model disorientation*, in fact, leads to randomly-initialized models not achieving their full potential (regardless of size), and discuss this effect in relation to how LLMs might be overcoming disorientation in applications.

## 2 SAFFU LAYER ARCHITECTURE

This derivation began by analyzing word2vec's continuous bag-of-words (CBOW) variant (Mikolov et al., 2013a;b), and was generalized to simple single-layer LMs, and then all feed-forward neural networks with arbitrary non-negative feature sets, as it is presented in **Appendix A**. Derived model-parameters are generally based on co-occurrences, requiring some re-normalization and non-linear transformation to approximate points of loss minimization. The discovery of the *priming number*—a constant dependent that allows conversion of input-output co-occurrence into well-optimized neural models—should not be understated, e.g., allowing extension of explicit solution applications from text (categorical) to image (numerical) input. Beyond extending explicit solutions to other data

types, discovering the priming number hinted at the possibility of *complex and multi-layer* solutions. Our work now picks up from that point, stacking multiple single-layer warm-starts to form multi-layer architectures, and further, investigates compositionally-bound layers and an encoder-decoder architecture combining self-attention and feed-forward layers wrapped in a generalized neural unit.

## 2.1 SELF-ATTENTIVE FEED-FORWARD NEURAL UNITS (SAFFUS)

We first define the data on which SAFFUs will operate, assuming sequential instances: a model's objective is to reconstruct a matrix $\boldsymbol{Y} \in \{0,1\}^{M \times N}$ of unit-normalized rows: $\|\boldsymbol{Y}_{m,:}\|_1 = 1$ corresponding to target elements for prediction. Predictions are based on $M$ sets of matrix-features contained in a tensor storing $K$ vectors of dimension $D$ for each $m = 1, \cdots, M$: $\mathbf{X} \in \mathbb{R}^{M \times K \times D}$. Thus, each $m$-target: $\boldsymbol{Y}_{m,:}$ has a slice from $\mathbf{X}_{m,:,:} \in \mathbb{R}^{K \times D}$ that is a matrix of $K$ vectors, drawn from other rows of $\boldsymbol{Y}$. LMs are auto-regressive, so each $m$-prediction has every $k = 1, \cdots K$ of its features drawn from an $i$-row of $\boldsymbol{Y}$: $\mathbf{X}_{m,k,:} = \boldsymbol{Y}_{i,:}$, or some low-dimensional embedding matrix, $\boldsymbol{E} \in \mathbb{R}^{N \times D}; D < N$.

Standard self-attention layers have a layer-specific dimension: $D_A$ and three parameter matrices: $\boldsymbol{W}_q, \boldsymbol{W}_k, \boldsymbol{W}_v \in \mathbb{R}^{D \times D_A}$; used together with the vector-valued *softmax* activation function: $\varphi(\boldsymbol{x})_i = e^{\boldsymbol{x}_i}/\sum_j e^{\boldsymbol{x}_j}$. Attention distributions: $\boldsymbol{A} \in \mathbb{R}^{N \times K}$ are applied for all $M$ predictions: $\boldsymbol{A} = \varphi(\mathbf{X}_{m,:,:}\boldsymbol{W}_q\boldsymbol{W}_k^T\mathbf{X}_{m,:,:}^T)$ to weight vectors for each $m$, producing hidden states: $\boldsymbol{H} = \boldsymbol{A}\mathbf{X}_{m,:,:}$ and score vectors: $\boldsymbol{H}\boldsymbol{W}_v$, the latter of which are passed through application-specific activation functions, such as the rectified linear unit (ReLU) (Fukushima, 1975; Nair & Hinton, 2010).

We first propose eliminating $D_A$. This is accomplished easily within $\varphi$, since the product $\boldsymbol{W} = \boldsymbol{W}_q\boldsymbol{W}_k^T \in \mathbb{R}^{D \times D}$ is equivalent to its component-wise formulation: $\boldsymbol{A} = \varphi(\mathbf{X}_{m,:,:}\boldsymbol{W}\mathbf{X}_{m,:,:}^T)$. This forces the re-consideration of $\boldsymbol{W}_v$'s use of $D_A$, which could instead be thought of as a hidden *or* decoder dimension, provided one defines $D_A = N$. We notate decoders by $\boldsymbol{U} \in \mathbb{R}^{D \times N}$, making the pre-activation form for a two-layer self-attention plus decoder model easily expressable as: $\varphi(\mathbf{X}_{m,:,:}\boldsymbol{W}\mathbf{X}_{m,:,:}^T)\mathbf{X}_{m,:,:}\boldsymbol{U}$. This standard matrix expression obfuscates the softmax function's input-output structure, but the attention layer operates *by-query*, i.e., $\varphi$ normalizes by *row*. If queries are defined by $h^{\text{th}}$ features, score vectors can be expressed individually as: $\varphi(\mathbf{X}_{m,h,:}\boldsymbol{W}\mathbf{X}_{m,:,:}^T)\mathbf{X}_{m,:,:}\boldsymbol{U}$.

We next ask if quadratic form for $\boldsymbol{A}_{m,:}$ be computed in a way separating $\mathbf{X}$ from $\boldsymbol{W}$, exchanging the order of self-attention's multiplication to: $\boldsymbol{A}_{m,:} = \varphi(\mathbf{X}_{m,h,:}\mathbf{X}_{m,:,:}^T\boldsymbol{W})$. Note that this formulation requires redefining the dimensionality of $\boldsymbol{W} \in \mathbb{R}^{K \times K}$. To concisely notate, we store *consolidated* quadratic features for each target in $\boldsymbol{Q} \in \mathbb{R}^{M \times K}$, defined by-$m$ as: $\boldsymbol{Q}_{m,:} = \mathbf{X}_{m,h,:}\mathbf{X}_{m,:,:}^T \in \mathbb{R}^K$, which refines the hidden-state equation to: $\boldsymbol{H}_{m,:} = \varphi(\boldsymbol{Q}_{m,:}\boldsymbol{W})\mathbf{X}_{m,:,:}$. Finally, we propose a negative logarithm operate on attention-outputs: $\boldsymbol{A}_{m,:} = -\log\varphi(\boldsymbol{Q}_{m,:}\boldsymbol{W})$. While the softmax operates on score vectors: $\varphi(\boldsymbol{A}_{i,:}\mathbf{X}_{m,:,:}\boldsymbol{U})$, attention's log-softmax *mathematically* 'activates' features by providing separation in differential structure between attention and decoder layers that makes a solution tractable. Queries from the layer's *head* $h$—a hyperparameter—are used to compute outputs:

$$\text{SAFFU}(\mathbf{X}_{m,:,:}) = \varphi(-[\log\varphi(\mathbf{X}_{m,h,:}\mathbf{X}_{m,:,:}^T\boldsymbol{W})]\mathbf{X}_{m,:,:}\boldsymbol{U}). \tag{1}$$

## 2.2 AN EXPLICIT FORM FOR FEED-FORWARD OPTIMIZATION

Motivation for log-probability activation becomes clearer when the explicit solution proofs are considered in **Appendices A** and **B**, where logits partly invert softmax operations. Proof requires defining hidden state vector-sums: $\boldsymbol{H}_{m,:} = \sum_{k=1}^{K} \mathbf{X}_{m,k,:}$, the decoder's action: $\hat{\boldsymbol{Y}}_{m,:} = \varphi(\boldsymbol{H}_{m,:}\boldsymbol{U})$, and:

**Definition**: A data set of vector-inputs $\boldsymbol{H} \in \mathbb{R}^{M \times D}$ and -outputs $\boldsymbol{Y} \in \mathbb{R}^{M \times N}$ has generalized co-occurrences $\boldsymbol{F}(\boldsymbol{H}, \boldsymbol{Y}) \in \mathbb{R}^{D \times N}$ between inputs and outputs defined by the sum of outer products:

$$\boldsymbol{F}(\boldsymbol{H}, \boldsymbol{Y}) = \sum_{m=1}^{M} \boldsymbol{H}_{m,:} \otimes \boldsymbol{Y}_{m,:} = \boldsymbol{H}^T\boldsymbol{Y}. \tag{2}$$

**Theorem**: A softmax-activated feed-forward layer receiving $K$-norm non-negative $D$-dimensional inputs $\boldsymbol{H}_{m,:}$ for each target of prediction $\boldsymbol{Y}_{m,:}$ is approximately optimized by a column-wise translation of the layer's generalized log-co-occurrence matrix: $\boldsymbol{U}_{j,i} = \log\boldsymbol{F}(\boldsymbol{H}, \boldsymbol{Y})_{j,i} + w_i$. The translating weights, $w_i$, are defined by $i$-column (output) as: $w_i = \frac{K-1}{K}\log(\sum_{d=1}^{D}\boldsymbol{F}(\boldsymbol{H}, \boldsymbol{Y})_{d,i})$, defining

an explicit form for each of the layer's $j, i$-parameters by the expression:

$$U_{j,i} = \log F(H, Y)_{j,i} - \frac{K-1}{K} \log \left( \sum_{d=1}^{D} F(H, Y)_{d,i} \right) \tag{3}$$

Proof of the above is recorded in **Appendix A**. We refer to $K$ as a *priming number*, and in circumstances where features are not unit-normalized (but still positive) the explicit solution appears to still function quite well. To extend the priming number from discrete feature sets, the average norm of a given feature vector: $\hat{K} = (\sum_{m=1}^{M} \sum_{d=1}^{D} H_{m,d})/M$ is effective. However, the most critical knowledge explicit solution use is *understanding layer inputs and targets*. Decoders—such as $U$ in the theorem—often have clear inputs (features) and outputs (supervising targets); however, *compositional* layers like $W$—within a SAFFU's 'deep' attention layer—require investigation to determine an answer to: *what supervises self-attention?*

### 2.3 Extending the explicit solution from single layers to SAFFUs

The explicit solution to single layers tells us in part that: first-order approximations can be computed locally from generalized log-co-occurrences matrices, from the bottom up. However, these kinds of local/first-order approximations are *non-compositional*, that is, even when they are applied to multi-layer softmax networks, their local optimization will is of lower quality than what's achievable by backpropagation, which utilizes the differential structure of function composition to tease higher-order behavior out of networks. We acknowledge this, specifically, to highlight that the SAFFU's explicit solution *is* the first such *compositional* explicit solution—our task is to train an LM by minimizing the cross entropy of SAFFU layers over $W$ and $U$:

$$L = - \sum_{m=1}^{M} \log \text{SAFFU}(\mathbf{X}_{m,:,:})_{i_m} = - \sum_{m=1}^{M} \log \varphi(-\log \varphi(W\mathbf{X}_{m,h,:}\mathbf{X}_{m,:,:})\mathbf{X}_{m,:,:}U)_{i_m} \tag{4}$$

where $i \in \{1, \cdots, N\}^M$ is the vector of target indices for each prediction in the sequence of $M$.

#### 2.3.1 Optimizing a SAFFU's decoder layer

Supposing one already possessed an optimized attention layer $W$, our notational conventions for the $M$ attention distributions: $A_{m,:} = -\log \varphi(W\mathbf{X}_{m,h,:}\mathbf{X}_{m,:,:})$ and their corresponding hidden states: $H_{m,:} = A_{m,:}\mathbf{X}_{m,:,:}$ make direct application of **Eq. 7** straightforward with knowledge of $U$'s priming number: $K_U$. The negative logarithm in $A$'s definition is not unit-normalized, but an an upper bound on its values—the negative logarithm of a probability distribution, i.e., entropy—is easily obtained from a uniform distribution: $K_U = K \log K$, recording the layer aggregation of $K$ unit-normalized features using $K$ entropically-activated probabilities as feature weights. With $K_U$, we can fully apply **Eq. 7** over $H$ and $Y$ to state $U$'s explicit form:

$$U_{j,i} = \log F(H, Y)_{j,i} - \frac{K_U - 1}{K_U} \log \left( \sum_{d=1}^{D} F(H, Y)_{d,i} \right) \tag{5}$$

Note that computing $F(H, Y)$ requires $W$ being known form first: $H_{m,:} = -\log \varphi(Q_{m,:}W)\mathbf{X}_{m,:,:}$, i.e., $U$'s explicit solution can only be computed *from $W$*.

#### 2.3.2 Optimizing a SAFFU's attention layer

**Appendix B** presents finer details on the derivation the SAFFU's explicit solution. This solution relies on direct application of **Eq. 7**, and requires answering the question: "*what supervises self-attention?*" One can think of self-attention as producing feature-weighting distributions, and perhaps could anticipate that supervising information for a self-attention distribution is 1) dependent on its decoder, and 2) guides weights to features that are most predictive of targets. Ultimately, solving $L$'s derivatives with respect to $W_{i,j}$ set equal to 0 lead us to the revelation that $V \in \mathbb{R}^{M \times K}$ defined by $V_{m,k} = [U_{:,i_m} - U\varphi(H_{m,:}U)] \cdot \mathbf{X}_{m,k,:}$ was 'supervising' $W$, i.e., as an analog to $Y$ (see **Appendix B.2**). While we intentionally consolidated the attention layer's inputs under the form $Q$, it was a *marvel*—whether by serendipity or the need for concise notation—that the matrix $V$ emerged. In it contains variational information about the decoder matrix $U$, which summarizes what the attention-matrix $W$ should expect from $U$'s reactions to its ($W$'s) activations.

By comparing the co-optimial criteria of $U$ and $W$ in **Eqs. 18 and 22**, we were able to state concretely that the input-output pair of matrices $Q$ and $V$ are to $W$, as the pair $H$ and $Y$ are to $U$ in **Appendix B.2**. However, there are some differences to note between **Eqs. 18 and 22**. In particular, while the decoder's softmax only engages one output dimension at a time in its derivative via $Y_{m,i}$ in **Eqs. 17–Eq. 18**, the attention layer's softmax has a derivative that engages *all* of its output dimensions simultaneously via $\sum_{k=1}^{K} V_{m,k}$ in **Eqs. 21–22**. Regardless, the matrix $V$ represents the "internal" targets of the SAFFU—supervising $W$ to temper its features to the decoder's variation—leaving $W$'s priming number $K_W$ as the only remaining unknown in its explicit solution:

$$W_{j,i} = \log F(Q,V)_{j,i} - \frac{K_W - 1}{K_W} \log\left(\sum_{k=1}^{K} F(Q,V)_{k,i}\right) \tag{6}$$

While estimating a 'good' value of $K_U$ depended on the input data in $\mathbf{X}$ *and* the functional form of the layer defined by $W$, $W$'s priming number, itself, depends *only* on its input features in $Q$. Consolidated quadratic features in $Q$ are defined as $Q_{m,:} = \mathbf{X}_{m,h,:}\mathbf{X}_{m,:,:} \in \mathbb{R}^K$, where each vector $Q_{m,:}$ contains $K$ inner products of the vector inputs from $\mathbf{X}_{m,:,:} \in \mathbb{R}^{K \times D}$ with their head-feature $h$. These are inner products between unit-normalized vectors, so their values $Q_{m,:}$ can be thought of as similarities between the head feature and the others in $\mathbf{X}_{m,:,:}$. Thus, while $Q$'s values are each less than one, one should expect $\|Q_{m,:}\|_1 > 1$. However, the norms of vectors in $Q$ are bounded: $\|Q_{m,:}\|_1 \in [0, K]$, since each 'similarity' cannot have value greater than 1. Thus, a sub-linear, increasing function of $K$ is likely useful for estimation of $W$'s priming number, and, we set $K_W$ at: $K_W = \log K$ for simplicity.[1] However, it's likely the case that for $K_W$ (and $K_U$) can be refined further by setting their values to the average norms of their input vectors. Finally, since computing $F(Q,V)$ requires knowledge of $U$ ($V$'s expression depends on $U$), we note that one must independently have *some* initial solution to *either* $U$ or $W$ before the other can be computed.

## 2.4 INITIALIZING SAFFUS

The co-dependence between the explicit solutions for $W$ and $U$ is a start-up problem, where one needs only a guess to get the process going. This could be a 'dumb' guess, like a uniform, e.g., all-1 initialization for $W$, or it could be more nuanced and estimate $W$ (or $U$), and perhaps alternatingly update their values until a stopping criterion is reached. For a non-uniform initial guess at $W$, one must consider the input data's distributional structure. The vectors contained within $\mathbf{X}$ will generally be word embeddings, and we require only that word embeddings are non-negative and unit-normalized. Standard word embeddings can be coerced to this domain via a variety of methods, e.g., by passing traditional vectors through a softmax function. Regardless, we denote emebedding layers by $E \in (0,1]^{N \times D}$, and assume that each $i$-token's embedding vector (from the vocabulary of $N$) has a unit 1-norm: $\|E_{i,:}\|_1 = 1$. Furthermore, embedding layers *with* the same hidden dimension as the decoder layer ($D$) can be transformed similarly to $V$ to grossly improve initialization of $W$ *over* uniform values: $\hat{V}_{m,k} = \left[\log E_{i_m,:} - \log E^T \cdot \sum_{j=1}^{M} \frac{Y_{j,:}}{M}\right] \cdot \mathbf{X}_{m,k,:}^T$. All testing with SAFFUs has demonstrated this initialization grossly-outperforms uniform starts, and accelerates optimization.

Finally, note first that *both* of **Eq. 5** and **Eq. 6** rely upon a logarithm of their generalized co-occurrence matrices. The explicit solution's expression for $U_{i,j}$ in **Eq. 5** has both targets and features which are by-definition positive-valued; however, the $V$-*targets* for the attention-matrix solution in **Eq. 6** will likely contain negative values, and subsequently, have the potential to introduce negatives into $F(Q,V)$. While the logarithm can be extended from $(0,\infty)$ to $\mathbb{C} \setminus \{0\}$, the explicit solution only applies to positive-valued co-occurrences.[2] Thus, we translate variational inputs by a pre-determined constant bound, $c = 2(1 + 1/K) \log N$, within the definitions: $V_{m,k} = [U_{:,i_m} - U\varphi(H_{m,:}U) + c] \cdot \mathbf{X}_{m,k,:}$ and $\hat{V}_{m,k} = \left[\log E_{i_m,:} - \log E^T \cdot \sum_{j=1}^{M} \frac{Y_{j,:}}{M} + c\right] \cdot \mathbf{X}_{m,k,:}^T$. The bound $c$ can be understood as 2—since $V$ is computed via differences of *two* vectors—times the product of the exponent derived from a model's priming number ($K$), with the maximum entropy from a uniform distribution over a vocabulary of size $N$, since the columns of $U$ approximately equal log-probabiltiy distributions. Computationally, $c$ appears to produce matrices $F(Q,V)$ woth positive values for all architectural variants tested. Intuitively, we understand the robustness of the SAFFU's explicit solution to $V$'s translation by $c$ (as defined), as a result of each vector in:

---

[1] Setting $K_W = \log K$ immediately improved performance over the value $K_W = K$ in early testing.
[2] Negative inputs require extension of $\varphi$ over a complex domain, which is beyond this work's scope.

$U_{m,:} - U\varphi(H_{m,:}U)$ being in the pre-image of the softmax function's prediction from a uniform feature-vector over the decoder $U$. Thus, the translation of each pre-image vector—and hence their difference—is an operation to which the softmax function input is invariant (scalar translation).

## 2.5 Assigning Low-dimensional Input Vectors (Embeddings)

Standard-basis encoding underlies token representation in neural language processing, even when tokens are mapped sparsely to low-dimensional embeddings. While standard bases are excellent for representation from perspectives such as precision, simplicity, and transparency, their relatively high dimensionalities make dimensionality reduction necessary—standard bases scale poorly and over-fit to training data, to name a few issues. Dimensionality reduction can be handled via gradient-based optimization, but this approach is largely antithetical to our work's approach. Thus, we employ a naïve mathematical approach, that 1) selects a low dimension: $D$ (a hyperparameter) and extends its set of standard basis vectors in the identity matrix: $I \in \{0,1\}^{D \times D}$, to a larger set of up to $2^D - 1$ *bit-vectors*, in order of decreasing discernability, to rapidly train embedding matrices of unit-normalized bit-vectors to satisfy the SAFFU's representation requirements for $E \in (0,1]^{N \times D}$. Pseudocode is presented in **Appendix C** for the *bit-cipher algorithm*, which is applied in our assignment of bit-vectors in SAFFU model embedding layers to tokens, as well as to the training of low-dimensional 'targets' to train hidden layers in our description of the encoder-decoder, SAFFU-based transformer architecture presented in the next section.

We likewise densify bit-vectors using a model of noise. This is done by computing a vector of token counts $f = \sum_{m=1}^{M} Y_{m,:}$, and then the average (un-noised) embedding: $\overline{e} = (\sum_{n=1}^{N} f_n E_{n,:})/M$, and a model: $q \in (0,1)^N$ for the portion of occurrences that each $n$-token's observations are (non-)erroneous. Assuming that the highest-count tokens are least erroneously observed, we assume that only one error is observed relative to each token's count, that is: $q_n = f_n/(f_n + 1)$. Next and regardless of the token that is observed, we modify its vector according to the probabilities that any different, $j$-token, should have been observed, instead, which will take the form of a normalized ($\|p\|_1 = 1$) *noise* vector: $p \in (0,1)^N$, defined to be near-uniform as: $p = (1 - \overline{e})/\|1 - \overline{e}\|_1$. To understand $p$ intuitively, we note that 1-minus each of the average embedding $\overline{v}$'s (normalized) valuea is also a probability, which expresses the chance that a given dimension's magnitude is spurious (should not be observed). In application, the value of each bit-vector, $E_{n,:}$, is finalized by adding noise to rows of embedding layers: $E_{n,:} = q_n E_{n,:} + (1 - q_n)p$.

## 3 A SAFFU-based Transformer Architecture

To define an LM and transformer architecture, we generally utilize two distinct SAFFUs, which are principally defined by hyperparameters referred to as the *block size*: $b$, and the *radius*: $r$. Both are positive integers greater than 1 that describe the number of features over which a SAFFU's attention layer operates. The block and radial SAFFUs utilize different definitions of context for input tensors, denoted by $\mathbf{X}^{\text{block}}$ and $\mathbf{X}^{\text{radius}}$. The value $b$ defines the number of tokens per input *block* for self-attention. Specifically, consider collecting a document's $M_{\text{doc}}$ tokens in the tensor $\mathbf{B} \in \mathbb{R}^{\lceil M_{\text{doc}}/(b-1) \rceil \times b \times D}$ by assigning each $m = 1, \cdots, M_{\text{doc}}$ to block $i = \lceil m/(b-1) \rceil$ by the equation $\mathbf{B}_{i,2:b,:} = [E_{i_j,:}]_{j=(i-1)(b-1)+1}^{i(b-1)+1}$. These input embeddings are broken into slices of $b - 1$ so as to accommodate room for special tokens that further contextualize input, by indicating if it is the first block or a later one. All blocks have their first input $\mathbf{B}_{i,1,:}$ set to an embedding for a *start of document* token: `"<sod>"` (for the first block), *or* to an embedding for a *fragment* token: `"<frg>"` (for other blocks). Padding tokens: `"<pad>"` fill the remaining positions of the last block with features, the last of which is reserved for an *end of document* token's: `"<eod>"` embedding.

Slices of the block-input tensor are assigned according to the equation: $\mathbf{X}_{m,:,:}^{\text{block}} = \mathbf{B}_{i,:,:}$. To assure that each slice $\mathbf{X}_{m,:,:}^{\text{block}}$ contains no target information ($Y_{m,:}$), inputs appearing at or beyond the target's position within the block are replaced by those for padding tokens. While $\mathbf{X}_{m,:,:}^{\text{block}}$ provides a *global* information on feature positions, the radius $r$ is *local*, i.e., has a sliding horizon of $r$ features for each target. Denote the $m^{\text{th}}$ target's position within block $i$ by $j$, and define the $r$-input radial features as those appearing before the $m^{\text{th}}$: $\mathbf{X}_{m,:,:}^{\text{radius}} = \mathbf{B}_{i,(j-1-r):(j-1),:}$ For targets at positions $m < r$ (without a complete radius), missing features are filled with `"pad"` embeddings. Each block and ra-

dius SAFFU can be operated under two modes of vector aggregation: summation-based aggregation (*sum*) models add attention-weighted input vectors, and concatenation-based (*cat*) models concatenate their attention-weighted input vectors. Note: *cat* models form hidden states in $\mathbb{R}^{KD}$ (vs. $\mathbb{R}^D$) and so incur a $K$-fold increase decoder-parametric complexity: $\boldsymbol{U}^{\text{cat}} \in \mathbb{R}^{KD \times N}$. This is controlled by setting separate embedding dimensions for each of the block and radial SAFFU's inputs: $D_b = 2^7$ and $D_r = 2^5$ for all experiments, [3], keeping our 'best' models under 10-million parameters.

For an encoder-decoder architecture, we require that both block and radial SAFFUs have their outputs reduced to a 'low' hidden dimension: $D_H < N$. This is accomplished by dimensionally reducing both block- and radius-SAFFU targets in explicit solutions from $\boldsymbol{Y}$ to the matrix $\tilde{\boldsymbol{Y}} \in \mathbb{R}^{M \times D_H}$ defined by: $\tilde{\boldsymbol{Y}}_{m,:} = \boldsymbol{Z}_{i_m,:}$. Here, $\boldsymbol{Z}$ is a matrix of bit-vectors—serving as low-dimensional/hidden targets—from the bit-cipher algorithm depicted in **Fig. 2**. Block and radial outputs are then be concatenated and decoded (again) by a final feed-forward layer: $\boldsymbol{M} \in \mathbb{R}^{2D_H \times N}$. A full architectural diagram for this design is presented at left in **Fig. 1**, where the top and bottom flows depict block and radial SAFFUs operating on sequentially ordered (from top to bottom), globally- and locally-positioned vectors (black rectangles). After products are taken with the head vector (depicted in yellow), quadratic features are passed to self-attention layers, which output positional weights (depicted in gray) to produce aggregate embeddings. Aggregates are fed through their decoders to produce concatenated outputs from the two SAFFUs, before being fed forward to the target distribution size. Thus, the last layer is the decoder, and all preceding layers comprise the encoder.

### 3.1 Augmenting Transformers with Document Models

To better contextualize a given transformer's outputs, we likewise define an optional *document model*, which outputs its own hidden state via a intermediate single-layer prediction. We assume that there are $\Delta$ documents, and that the $m^{\text{th}}$ token in document $\delta$ of length $M_\delta$ has its input to the document model defined by the average of all preceding embeddings (plus one for a padding token): $\boldsymbol{x} = \left(\boldsymbol{E}_{i_{\text{"<pad>"}},:} + \sum_{j=1}^{m-1} \boldsymbol{E}_{i_j,:}\right)/m$. Each vector $\boldsymbol{x}$ is passed through a feed-forward model whose parameter matrix we denote by $\boldsymbol{D} \in \mathbb{R}^{D \times \Delta}$ that predicts the document index $\delta$ from which $\boldsymbol{x}$ came. When a document model is utilized with a SAFFU-based transformer, each of its outputs: $\varphi(\boldsymbol{x}\boldsymbol{D})$ is concatenated to the result from the two SAFFU's, i.e., $\varphi(\boldsymbol{x}\boldsymbol{D})$ is concatenated to the red-blue result prior to the last feed-forward layer, $\boldsymbol{M}$, whose input dimensionality is augmented to: $\mathbb{R}^{(2D_H + \Delta) \times N}$.

## 4 Computational Experiments

**Data**. We perform all ablation—and other, larger experiments—on a recently-released data set, known as the BabyLM data set (Warstadt et al., 2023). These data have two main training sets, consisting of 10- (10M) and 100-million (100M) tokens, and likewise contain 10-million token sets for development and testing. For speed and efficiency, our ablation used the first $10\%$ (roughly $10^6$ tokens) of the 10M training set.

**Tokenization**. We use sub-word tokenizations to benefit from the efficiency, simplicity, and speed of a count-based implementation of byte-pair encoding (BPE) (Sennrich et al., 2016). We train two BPE models over the $2^{17}$ and $2^{20}$ highest-count words contained in the 10M- and 100M-word BabyLM data sets, respectively until the stopping condition: *all new merge rules produce a new sub-word token of count 1* is reached. *All* experiments had their vocabulary size further reduced by replacing sub-word tokens *not* needed for tokenization of the the $2^{12}$ highest count words. This reduced the 10M-token sub-word vocabulary size to a functional set of $N^{\text{10M}} = 2,848$ (down from $26,693$) sub-word tokens, which added large efficieny boosts to ablation time. However, we note that these ablation efficiency boosts were only achieved during backpropagation, since computing explicit solutions doesn't require operation of the final softmax, which is bottlenecked by a normalization over the vocabulary size $N$. The 100M-token model's vocabulary was also reduced, but from $20,590$ to $N^{\text{100M}} = 2,755$, which thus demonstrated a much higher compression ratio over its $2^{20}$ words, when compared to the 10M model's same-sized covering by $2,755$ sub-words, down from $20,590$.

---

[3]Early experimentation uniformly demonstrated that bit-cipher embeddings smoothly offset performance with size, which—alongside the clear 'best' configuration of *sum*-based block and *cat*-based radius aggregation—meant computational gains could be made by lowering parameter-intensive *cat* dimensions.

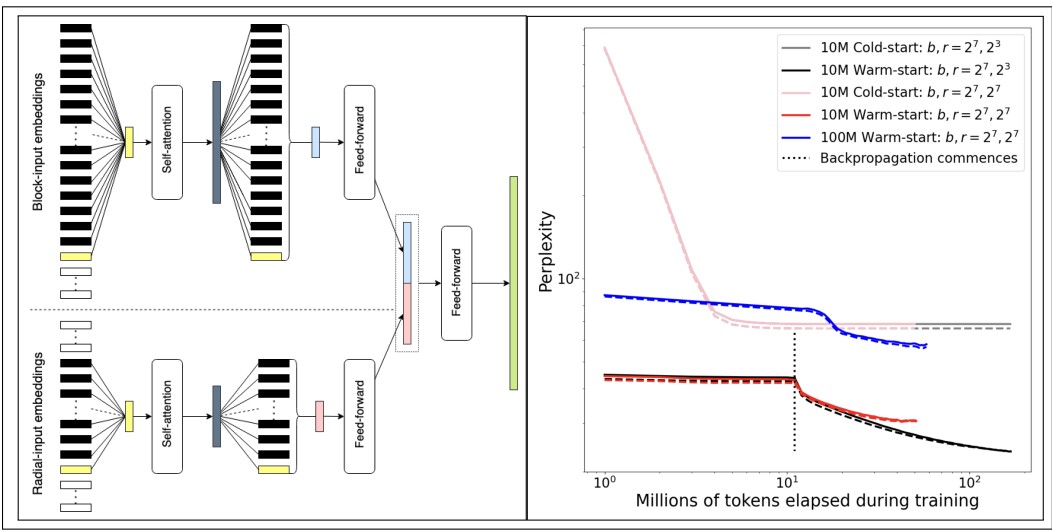

Figure 1: Cold-start (Cold train/dev) curves were obtained via backpropagation on randomly initialized parameters; whereas Warm-start (Warm train/dev) curves were obtained by first tuning the model with its explicit solution, and then applying backpropagation.

**Training**. Experiments were trained over 1-million token folds of the 10M- and 100M-token sets. Backpropagation experiments used Adam (Kingma & Ba, 2015) for optimization with a learning rate of $10^5/2$ across experiments. Ablation experiments utilize absolutely no backpropagation, and received only $10\%$ of the 10M-token data via *initialization*, defined as: having embedding matrices initialized by the bit-cipher algorithm, self-attention matrices initialized by the explicit solution *initialization* targets, $\hat{V}$, and then followed by successive application of explicit solution computations to all subsequent feed-forward/decoder layers, from the bottom up. In larger experiments, we refer to *cold-start* models as those which have had random parameter initialization followed by backpropagation applied to all layers. Cold-starts are compared to *warm-start* models, which have have initialization by $\hat{V}$ (on the first fold) followed by *tuning*. We distinguish tuning from initialization only by use of $V$ over $\hat{V}$. Tuning is applied over 10 folds (10-million tokens) for both 10M- and 100M-token models. While the 10M/former models utilized $V$ instead of $\hat{V}$ over 10 iterations, the 100M-token model was initialized in a single 10-million token shot, i.e., it was *only* initialize with $10\%$ of the larger data set before backpropagation. Following 10-million tokens warm-start, backpropagation was applied to all *but* the embedding layers of warm-start models, until early stopping is signaled by $2^3$ increases in perplexity, which was measured on approximately $10^5$ tokens from the development set, regardless of model size. Early stopping determines the total number of cold-start epochs, and we refer to non-altered bit-cipher embeddings as *frozen*, whose results are discussed in the next section. Abbreviated training logs from this process are provided in **Appendix 6**.

## 4.1 EXPERIMENTAL RESULTS

The explicit solution's efficiency and stability allowed ablation of many SAFFU model variants. All—approximately 250—have their performance presented in **Appendix D**. These explore combinations of the proposed *sum* and *cat* architectural variations on each of the block and radial SAFFUs (**Tabs. 1–4**), and then the impact of the document model on top of the 'best' combination (with lowest-perplexity models), which turned out to use *sum* for blocks and *cat* for radii (**Tab. 3**). Each table represents an $r - b$ 'grid' corresponding to powers of 2, i.e., with $r, b \in \{2^1, 2^2, \cdots, 2^7\}$. The tables in **Appendix D** can be seen as a basis for determination of which architectural variants merited further training. For planning larger-scale models, it is critical to observe that perplexities more or less generally decrease with larger values of $r$ and $b$ across tables, as this indicates that adding more features improves prediction. However, we note some local optima appear for smaller values of $r$ when its 'best' *cat*-based aggregation is utilized, providing a balance of efficiency and performance. While *cat*-based block aggregation is less advantageous, we note that it likewise has worse optima.

The 'best' architecture from **Tab. 5** (the black curve in **Fig. 1**), at the high-effiency optimum of $r = 2^3$ kept blocks large: $b = 2^7$, still capture long-range correlations. Setting $r = 2^3$ ultimately resulted in models with more robust learning curves, optimizing for more epochs before the early-stopping criterion was reached than when $r = 2^7$ (**Fig. 1**, red curve). Aside from ablation successfully guiding model experimentation, it is perhaps the biggest surprise to see that cold-start models *fail* to optimize to anywhere near the level of performance that warm-start models do, as can be seen in the gray and pink curves in **Fig. 1**. While it is perhaps not surprising that fewer parameters contributed to greater robustness during backpropagation, there would likely have been no impetus to investigate the more-performant (and efficient) $r = 2^3$ model if our experiment not identified the near-parity between $r = 2^3$ with $r = 2^7$ in ablation. Ultimately, $r = 2^3$ achieved the best test perplexity of $23.84$, while the $r = 2^7$ model's perplexity only fell to $30.35$ and the 10M cold-start models reached $63.98$ (both), and our initial 100-million token model with $r = 2^3$ and $b = 2^7$, suprisingly, stopped at $58.05$ (blue in **Fig. 1**), despite having been trained on the most individual documents.

## 5    DISCUSSION

Ablation-based determination of 'best' models for backpropagation greatly benefited from using few tokens, which was possible due to the deterministic nature of explicit solutions and their initialization by zero-matrices. Tuning models beyond ablation improved performance; however, initializing over just 1-million tokens with the explicit solution demonstrated balanced performance on a random development set. For the 100-million token model (blue in **Fig. 1**), this motivated a simplified training process that applied backpropagation immediately after its initialization over *10-million* tokens in a single pass. These $10\%$ of the 100M model's training data appeared insufficient for warming the model up to learning from all 100M tokens. This resulted in demonstrably less stable backpropagation, when compared to the faster-to-optimize 10M-models. Hence, having a broader 'foundation' with more data used in the explicit solution could be key to stable learning and generalization.

Several new algorithms were required to satisfy the strict conditions defined by our work, but, when taken as a whole—with this paper's computational experiments demonstrating warm-start optimality unachievable *without* explicit solutions—this work shows that when iterative optimization and explicit solutions are combined, they can lead to neural optimizations that were previously un-achievable over such marginal scales of data. The derived explicit solutions at present only work *well*, meaning that while they drastically reduce the training costs traing networks, they still must be followed by some iterative optimization using backpropagation. In other words, these solutions are *not perfect*. Though this is a limitation, awareness to it directs further development towards the possibility of future work on explicit solutions that continues to reduce the expense of training networks by fully eliminating the need for backpropagation. Regardless, the presented explicit solutions *do* make it possible to optimize a given network's performance to a point that is far beyond what's possible for networks with parameters that were initialized randomly. *Critically*, we note that this effect is present regardless of how much data is used for training.

## 6    CONCLUSION

While training bigger models, we noted how the 10M data—which was fully used in its explicit solution applications—was better used during backpropagation before early stopping, optimizing models more effectively over multiples passes of relatively few data. Alongside this smoother optimization, what's truly striking about is that training over multiple passes on small samples of data was more effective than access to more data (for fewer passes). So, while one might expect the 100M-token data to produce better models, it may be the case that they needs be trained over for longer periods, and perhaps have the explicit solution utilized over all 100M tokens. Understanding this phenomenon further will be critical for future applications of the SAFFU architecture and its explicit solution, but the best scenario for future development is likely if explicit solutions can be derived that entirely obviate the need for backpropagation. Regardless, since explicit solutions seem to work well with less data, they hold potential for the future development of high-performance LMs that moreover, are *small*, and which could learn on-site using localized data for applications of embodiment. These findings demonstrate the potential of explicit solutions for efficiently training complex and multi-layer models, and we hope this work encourages further exploration of explicit solutions as a strategy for improving training efficiency and our understanding of model function.

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

## A  DERIVATION OF THE SINGLE-LAYER MODEL'S EXPLICIT SOLUTION

**Theorem**: A softmax-activated feed-forward layer receiving $K$-norm non-negative $D$-dimensional inputs $\boldsymbol{H}_{m,:}$ for each target of prediction $\boldsymbol{Y}_{m,:}$ is approximately optimized by a column-wise translation of the layer's generalized log-co-occurrence matrix: $\boldsymbol{U}_{j,i} = \log \boldsymbol{F}(\boldsymbol{H},\boldsymbol{Y})_{j,i} + \log \boldsymbol{w}_i$. The translating weights, $\log \boldsymbol{w}_i$, are defined by $i$-column (output) as: $\log \boldsymbol{w}_i = \frac{K-1}{K} \log(\sum_{d=1}^{D} \boldsymbol{F}(\boldsymbol{H},\boldsymbol{Y})_{d,i})$, defining an explicit form for each of the layer's $j,i$-parameters by the expression:

$$\boldsymbol{U}_{j,i} = \log \boldsymbol{F}(\boldsymbol{H},\boldsymbol{Y})_{j,i} - \frac{K-1}{K} \log \left( \sum_{d=1}^{D} \boldsymbol{F}(\boldsymbol{H},\boldsymbol{Y})_{d,i} \right) \tag{7}$$

**Proof**: Abbreviating $\boldsymbol{F}(\boldsymbol{H},\boldsymbol{Y})$ by simply $\boldsymbol{F}$ for concise notation, first re-arrange the starting expression for a $j,i$-index pair of $\boldsymbol{U}$ is:

$$\boldsymbol{U}_{j,i} = \log \boldsymbol{w}_i \boldsymbol{F}_{j,i} \tag{8}$$

It is a matter of algebra to reduce the the likelihood function's general form with $\boldsymbol{w}$ to an expression only dependent on $\boldsymbol{w}$'s components in the denominator-factors:

$$e^L = \prod_{m=1}^{M} \frac{e^{\boldsymbol{H}_{m,:}\boldsymbol{U}_{:,\boldsymbol{i}_m}}}{\sum_{n=1}^{N} e^{\boldsymbol{H}_{m,:}\boldsymbol{U}_{:,n}}} = \prod_{m=1}^{M} \frac{\prod_{d=1}^{D} \boldsymbol{F}_{d,\boldsymbol{i}_m}^{\boldsymbol{H}_{m,d}}}{\boldsymbol{w}_{\boldsymbol{i}_m}^{-K} \sum_{n=1}^{N} \boldsymbol{w}_n^K \prod_{d=1}^{D} \boldsymbol{F}_{d,n}^{\boldsymbol{H}_{m,d}}} \tag{9}$$

Above, the expression shows that one need only minimize the denominator at right to maximize the overall expression, i.e., optimize the likelihood. Since the logarithm is a monotone function, this is likewise equivalent to maximizing the logarithm of the denominator, which we denote by $\Upsilon$:

$$\Upsilon = \sum_{m=1}^{M} \epsilon_m = \sum_{m=1}^{M} \log\left[ \boldsymbol{w}_{\boldsymbol{i}_m}^{-K} \sum_{n=1}^{N} \boldsymbol{w}_n^K \prod_{d=1}^{D} \boldsymbol{F}_{d,n}^{\boldsymbol{H}_{m,d}} \right] \tag{10}$$

We then proceed directly, by differentially optimizing $\Upsilon$ and compute partial derivatives of $\epsilon_m$:

$$\frac{\partial \epsilon_m}{\partial \boldsymbol{w}_i}\bigg|_{\boldsymbol{i}_m=i} = \frac{K\boldsymbol{w}_i^{K-1} \prod_{d=1}^{D} \boldsymbol{F}_{d,i}^{\boldsymbol{H}_{m,d}}}{\sum_{n=1}^{N} \boldsymbol{w}_n^K \prod_{d=1}^{D} \boldsymbol{F}_{d,n}^{\boldsymbol{H}_{m,d}}} - \frac{K}{\boldsymbol{w}_i} \quad ; \quad \frac{\partial \epsilon_m}{\partial \boldsymbol{w}_i}\bigg|_{\boldsymbol{i}_m \neq i} = \frac{K\boldsymbol{w}_i^{K-1} \prod_{d=1}^{D} \boldsymbol{F}_{d,i}^{\boldsymbol{H}_{m,d}}}{\sum_{n=1}^{N} \boldsymbol{w}_n \prod_{d=1}^{D} \boldsymbol{F}_{d,n}^{\boldsymbol{H}_{m,d}}} \tag{11}$$

Putting these pieces together in-sum produces the expression:

$$\frac{\partial \Upsilon}{\partial \boldsymbol{w}_i} = -\frac{K\boldsymbol{f}_i}{\boldsymbol{w}_i} + \sum_{m=1}^{M} \frac{K\boldsymbol{w}_i^{K-1} \prod_{d=1}^{D} \boldsymbol{F}_{d,i}^{\boldsymbol{H}_{m,d}}}{\sum_{n=1}^{N} \boldsymbol{w}_n \prod_{d=1}^{D} \boldsymbol{F}_{d,n}^{\boldsymbol{H}_{m,d}}} \tag{12}$$

it becomes helpful now to identify a weighted, geometric mean of co-occurrences with token $i$ over the $m^{\text{th}}$ instance's features: $\mathbb{E}_G[\boldsymbol{F}_{j,i} \mid \boldsymbol{H}_{m,:}] = (\prod_{d=1}^{D} F_{d,i}^{\boldsymbol{H}_{m,d}})^{1/K}$. Provided their inner product with the weights is approximately a constant $c \in \mathbb{R}$:

$$\sum_{n=1}^{N} \boldsymbol{w}_n \mathbb{E}_G[\boldsymbol{F}_{j,n} \mid \boldsymbol{H}_{m,:}] \approx c, \tag{13}$$

solving for $\frac{\partial \Upsilon}{\partial \boldsymbol{w}_i} = 0$ results in the following proportionality for each token-index, $i$:

$$\frac{\boldsymbol{f}_i}{\boldsymbol{w}_i^K} = \sum_{m=1}^{M} \frac{\mathbb{E}_G[\boldsymbol{F}_{j,i} \mid \boldsymbol{H}_{m,:}]^K}{\sum_{n=1}^{N} \boldsymbol{w}_n \mathbb{E}_G[\boldsymbol{F}_{j,n} \mid \boldsymbol{H}_{m,:}]^K} \propto \sum_{m=1}^{M} \mathbb{E}_G[\boldsymbol{F}_{j,i} \mid \boldsymbol{H}_{m,:}]^K \tag{14}$$

Each $\mathbb{E}_G[\boldsymbol{F}_{j,i} \mid \boldsymbol{H}_{m,:}]$ likely correlates to $\boldsymbol{f}_i$, and their sum further integrates a broader average:

$$\sum_{m=1}^{M} \mathbb{E}_G[\boldsymbol{F}_{j,i} \mid \boldsymbol{H}_{m,:}]^K = M\langle \mathbb{E}_G[\boldsymbol{F}_{j,i} \mid \boldsymbol{H}_{m,:}]\rangle^K \tag{15}$$

Here, the expression $\langle \mathbb{E}_G[\boldsymbol{F}_{j,i} \mid \boldsymbol{H}_{m,:}]\rangle$ indicates the $K$-power mean *of* the geometric means of co-occurrences with token $i$. We thus find that an explicit form for $\boldsymbol{w}_i$'s proportionality is:

$$\boldsymbol{w}_i \propto \frac{\boldsymbol{f}_i^{1/K}}{\langle \mathbb{E}_G[\boldsymbol{F}_{j,i} \mid \boldsymbol{H}_{m,:}]\rangle} \propto \boldsymbol{f}_i^{\frac{1-K}{K}} = \left( \sum_{d=1}^{D} \boldsymbol{F}(\boldsymbol{H},\boldsymbol{Y})_{d,i} \right)^{\frac{1-K}{K}}, \tag{16}$$

dependent on the double-averaged denominators scaling with count: $\langle \mathbb{E}_G[\boldsymbol{F}_{j,i} \mid \boldsymbol{H}_{m,:}]\rangle \propto \boldsymbol{f}_i$. ∎

## B  DERIVING THE SAFFU'S OPTIMIZATION CRITERIA

### B.1  DERIVING AN OPTIMIZATION CRITERION FOR THE DECODER LAYER

Consider the partial derivatives of $L$ with respect to each parameter of the decoder layer, $\boldsymbol{U}_{j,i}$:

$$\frac{\partial L}{\partial \boldsymbol{U}_{j,i}} = \sum_{m=1}^{M} \left[ \boldsymbol{Y}_{m,i} - \varphi\left(\boldsymbol{H}_{m,:}\boldsymbol{U}\right)_i \right] \boldsymbol{H}_{m,j} = \boldsymbol{F}(\boldsymbol{H},\boldsymbol{Y})_{j,i} - \sum_{m=1}^{M} \varphi\left(\boldsymbol{H}_{m,:}\boldsymbol{U}\right)_i \boldsymbol{H}_{m,j} \tag{17}$$

To derive these, it is helpful to recall the SAFFU's notational conventions (slices of matrices) for its $M$ attention distributions: $\boldsymbol{A}_{m,:} = -\log\varphi(\boldsymbol{W}\mathbf{X}_{m,h,:}\mathbf{X}_{m,:,:})$ and hidden states: $\boldsymbol{H}_{m,:} = \boldsymbol{A}_{m,:}\mathbf{X}_{m,:,:}$.

Upon setting the derivative equal to $0$, we then note that any optimum of the decoder must have its action over hidden states produce probabilities with expected values equal to the column-conditional probabilities of the co-occurrence distribution normalized by $\sum_{n=1}^{N} \boldsymbol{F}(\boldsymbol{H},\boldsymbol{Y})_{j,n} = \sum_{n=1}^{M} \boldsymbol{H}_{n,j}$:

$$\frac{\boldsymbol{F}(\boldsymbol{H},\boldsymbol{Y})_{j,i}}{\sum_{n=1}^{N} \boldsymbol{F}(\boldsymbol{H},\boldsymbol{Y})_{j,n}} = \sum_{m=1}^{M} \varphi\left(\boldsymbol{H}_{m,:}\boldsymbol{U}\right)_i \frac{\boldsymbol{H}_{m,j}}{\sum_{n=1}^{N} \boldsymbol{F}(\boldsymbol{H},\boldsymbol{Y})_{j,n}} = \mathbb{E}\varphi\left(\boldsymbol{H}_{m,:}\boldsymbol{U}\right)_i \tag{18}$$

In other words, we arrive at the same conclusion about the compositional decoder as was made from analysis of the single-layer decoder: the arithmetic average prediction of the model on target $i$, taken over the $j$-hidden frequencies, will *behave* as the conditional probability of the outputs, given the inputs. With this is established, we must ask the question: what distributional form will in the inputs take? This is partly answered by taking derivatives with respect to the 'deep' (attention) layer.

## B.2 DERIVING AN OPTIMIZATION CRITERION FOR THE SELF-ATTENTION LAYER

Once can similarly take the partial derivatives with respect to the attention layer's, parameters $\boldsymbol{W}_{j,i}$:

$$\frac{\partial L}{\partial \boldsymbol{W}_{j,i}} = \sum_{m=1}^{M} \left[ \boldsymbol{U}_{:,\boldsymbol{i}_m} - \sum_{n=1}^{N} \boldsymbol{U}_{:,n} \varphi \left( \boldsymbol{H}_{m,:} \boldsymbol{U} \right)_n \right] \cdot \frac{\partial \boldsymbol{H}_{m,:}}{\partial \boldsymbol{W}_{j,i}} \tag{19}$$

The compositional relationship between the decoder and attention layers is defined by an inner product of hidden state partial derivatives with a decoder-based expression (that we will return to shortly). Note: this relationship is invariant to activation functions, and would be expressed in precisely the same manner for compositional optimization of 'deep' layers. The hidden-state vectors produce partials derivatives of vectors, too, when $L$'s partial is taken over the parameter $\boldsymbol{W}_{j,i}$:

$$\frac{\partial \boldsymbol{H}_{m,:}}{\partial \boldsymbol{W}_{j,i}} = \sum_{k=1}^{K} \boldsymbol{\mathsf{X}}_{m,k,:} \boldsymbol{Q}_{m,j} \left[ e_k^{(i)} - \varphi(\boldsymbol{Q}_{m,:} \boldsymbol{W})_i \right] \tag{20}$$

where $e^{(i)}$ is the standard basis vector of dimension $K$ with 1 at position $i$. To express a solution for $\boldsymbol{W}$, it next helps to define the tensor $\boldsymbol{V} \in \mathbb{R}^{M \times K}$, containing the information from variational vectors that describe the relative sensitivity of the decoder layer to each $m^{\text{th}}$ instance of the training set's $k = 1, \cdots, K$ features in $\boldsymbol{\mathsf{X}}$: $\boldsymbol{V}_{m,k} = \left[ \boldsymbol{U}_{:,\boldsymbol{i}_m} - \boldsymbol{U} \varphi \left( \boldsymbol{H}_{m,:} \boldsymbol{U} \right) \right] \cdot \boldsymbol{\mathsf{X}}_{m,k,:}$. Combining $\boldsymbol{V}$ with **Eq. 19** and **Eq. 20** simplifies the attention layer's partial derivatives:

$$\frac{\partial L}{\partial \boldsymbol{W}_{j,i}} = \sum_{m=1}^{M} \boldsymbol{Q}_{m,j} \left[ \boldsymbol{V}_{m,i} - \varphi(\boldsymbol{Q}_{m,:} \boldsymbol{W})_i \sum_{k=1}^{K} \boldsymbol{V}_{m,k} \right] \tag{21}$$

Thus, solving $\partial L / \partial \boldsymbol{W}_{j,i} = 0$ allows for terms to be re-arranged and a—surprisingly familiar—condition on the point of the *attention* layer's optimization to be resolved:

$$\frac{\boldsymbol{F}(\boldsymbol{Q}, \boldsymbol{V})_{j,i}}{\sum_{k=1}^{K} \boldsymbol{F}(\boldsymbol{Q}, \boldsymbol{V})_{j,k}} = \sum_{m=1}^{M} \varphi \left( \boldsymbol{Q}_{m,:} \boldsymbol{W} \right)_i \frac{\boldsymbol{Q}_{m,j} \sum_{k=1}^{K} \boldsymbol{V}_{m,k}}{\sum_{k=1}^{K} \boldsymbol{F}(\boldsymbol{Q}, \boldsymbol{V})_{j,k}} = \mathbb{E}\varphi \left( \boldsymbol{Q}_{m,:} \boldsymbol{W} \right)_i \tag{22}$$

Specifically, *if* (a big if) the values $\left[ \boldsymbol{Q}_{m,j} \sum_{k=1}^{K} \boldsymbol{V}_{m,k} \right] / \left[ \sum_{k=1}^{K} \boldsymbol{F}(\boldsymbol{Q}, \boldsymbol{V})_{j,k} \right]$ constitute a probability mass function over the training set's $M$ instances of prediction, our conclusion is a statement much like for the SAFFU's decoder, where we observed an equivalent premise to the single-layer model's explicit solution: the arithmetic average prediction—now a feature weight—of the *attention* layer on feature $i$, taken over the $j$-input co-frequencies with quadratic values in $\boldsymbol{Q}$, will *behave* as the conditional probability of the outputs, given the inputs. As mentioned for the decoder, we must ask: what distributional form will in the inputs take? It should now be clear that neither the inputs, $\boldsymbol{H}_{m,:}$, for the decoder—nor the inputs, $\boldsymbol{Q}_{m,:}$, to the attention layer—have unit sum. Thus, estimates will need to be made for both layers' priming numbers in order to complete their optimizations.

Note that this investigation glosses over a subtle and perhaps exciting point of observation—that we can now answer: *what self-attention's supervising targets are.* While we intentionally consolidated the attention layer's inputs under the form $\boldsymbol{Q}$, it is a *marvel*—whether by serendipity or the need for concise notation—that the matrix $\boldsymbol{V}$ emerged. Intuitively, it contains variational information on the decoder summarizing what the attention-matrix $\boldsymbol{W}$ should expect from $\boldsymbol{U}$'s reactions to its ($\boldsymbol{W}$'s) activations. In summary, the co-optimal criteria for the matrices $\boldsymbol{U}$ and $\boldsymbol{W}$ in **Eqs. 18** and **22** ultimately allow us to see that $\boldsymbol{V}$ contains the differentially-guiding *targets* of the self-attention layer, $\boldsymbol{W}$. Thus, the input-output pair of matrices $\boldsymbol{Q}$ and $\boldsymbol{V}$ are to $\boldsymbol{W}$, as $\boldsymbol{H}$ and $\boldsymbol{Y}$ are to $\boldsymbol{U}$.

```
 1: procedure BIT-CIPHER(N, D)                    ▷ Construct a D-bit cipher of N ≤ 2^D dimensions.
 2:     B^(0) ← [0⃗]
 3:     for d = 1, ⋯, D do                        ▷ 1. Initialize lists for differently-normed bit-vectors.
 4:         B^(d) ← []
 5:     I ← Identity(b)
 6:     Z, E ← {0}^{N×D}, {0}^{N×D}
 7:     i, j, d ← 0, 0, 1
 8:     for n = 1, ⋯, N do
 9:         while E_{n,:} = 0⃗ do                  ▷ 2. Find the next norm-d (or d + 1) bit-vector.
10:             z ← Abs(B_j^{(d-1)} − I_{i,:})
11:             if ‖z‖_1 = d and z ∉ B^(d) then   ▷ 3. The norm must be d and the vector unused.
12:                 B^(d) ← Concatenate(B^(d), [z])
13:                 E_{n,:} ← z/‖z‖_1              ▷ 4. Normalize the bit-vector and assign as embedding.
14:                 Z_{n,:} ← z
15:             j ← j + 1
16:             if j = |B^{(d-1)}| then           ▷ 5. Change basis vector/component of modification.
17:                 j ← 0
18:                 i ← i + 1
19:                 if i = d then                 ▷ 6. Reverse the d-bit vector order and increment d.
20:                     if d = 1 then
21:                         I ← Reverse(I)
22:                     i ← 0
23:                     B^(d) ← Reverse(B^(k))
24:                     d ← d + 1
25:     return Z, E                               ▷ 7. Return matrices for deciphering and enciphering.
```

Figure 2: Bit-Cipher algorithm. After 1) initialization, the algorithm 2) finds new bit-vectors in decreasing order of discernability, by 3) identifying (unassigned) bit-vectors of increasing norm via translations of $d-1$-bit vectors by standard basis vectors. Unassigned bit-vectors are then 4) normalized and assigned as embeddings in $E$, while the raw bit-vectors, themselves are retained in $Z$ for training $D$-dimensional "hidden" states. Whenever the collection of $d-1$-bit vectors no longer has any unassigned $i$-component modifications, 5) the basis vector/component of modification must be incremented, and when this is the case for all last-component modifications, it's determined that there are no unassigned $k$-bit vectors, necessitating a 6) reversal of the $d$-bit vector order, which maintains smooth transitions of discernability, upon future assignment. 7) Once all $N$ dimensions have been assigned a bit-vector (and normalized counterpart), the $Z$ and $E$ are returned.

## C  BIT-CIPHER ALGORITHM DETAILS

To define the bit-cipher algorithm (depicted in **Fig. 2**), consider a vocabulary of $N$ unique tokens and select a 'low' dimension: $D \leq N$. Each $n^{\text{th}}$ token will have a unique bit vector assigned to its row in a matrix $Z \in \{0,1\}^{N×D}$ that is drawn from the larger collection of $2^N - 1$ non-zero bit vectors in $\{0,1\}^D$. The order of assignment from $\{0,1\}^D$ is based on a distinguishability hypothesis, which expects that a 'good' order increases vector norms, while assigning bit-vectors to more common categories (tokens). To assign bit-vectors in a 'smooth' order, the process depicted in **Fig. 2**, which starts at bit-vector norm $d = 1$ and inducts the order that $i = 1$: assigns standard basis vectors to the first $D$ rows of $Z$ from the identity $I$ to represent the $D$ most frequent tokens (generalizing one-hots/standard bases); $i = 2$: increments $d$ and adds standard-basis vectors from $I$ to those already assigned of norm $d - 1$ in $Z$ in reverse order of assignment, while filtering for unique bit-vectors in $\{0,1\}^D$; $i = 3$: repeats step $i = 2$. $D$-bit vectors are then normalized to meet SAFFU input requirements for its embedding layer, $E \in \mathbb{R}^{N×D}$, for each $n = 1, \cdots, N$: $E_{n:} = Z_{n:}/\|Z_{n:}\|_1$.

## D  ABLATION TABLES

Table 1: Training and Development-set perplexities **without a document model**. Aggregation modes are: **radial summation** and and **block summation**. Top-performing models are in **bold**.

|  | $b = 2^1$ | $b = 2^2$ | $b = 2^3$ | $b = 2^4$ | $b = 2^5$ | $b = 2^6$ | $b = 2^7$ |
|---|---|---|---|---|---|---|---|
| $r = 2^1$ | 61.8, 64.0 | 59.8, 62.0 | 59.2, 61.3 | 56.9, 58.9 | 55.2, 57.3 | 54.1, 56.1 | 53.4, 55.4 |
| $r = 2^2$ | 57.5, 59.5 | 56.5, 58.6 | 56.0, 58.1 | 54.0, 55.9 | 52.5, 54.5 | 51.6, 53.5 | 51.0, 52.9 |
| $r = 2^3$ | 56.1, 58.2 | 54.8, 56.8 | 54.6, 56.6 | 52.8, 54.7 | 51.5, 53.4 | 50.7, 52.6 | **50.2, 52.1** |
| $r = 2^4$ | 55.8, 58.0 | 54.5, 56.6 | 54.3, 56.4 | 52.7, 54.7 | 51.6, 53.6 | 50.9, 52.8 | 50.4, 52.3 |
| $r = 2^5$ | 56.8, 59.1 | 55.1, 57.4 | 55.1, 57.3 | 53.6, 55.6 | 52.4, 54.5 | 51.7, 53.7 | 51.3, 53.2 |
| $r = 2^6$ | 56.3, 58.6 | 55.2, 57.4 | 55.4, 57.6 | 53.8, 55.8 | 52.6, 54.6 | 51.9, 53.8 | 51.4, 53.3 |
| $r = 2^7$ | 57.1, 59.3 | 55.7, 57.9 | 55.5, 57.7 | 53.9, 55.9 | 52.7, 54.7 | 52.0, 53.9 | 51.5, 53.4 |

Table 2: Training and Development-set perplexities **without a document model**. Aggregation modes are: **radial summation** and **block concatenation**. Top-performing models are in **bold**.

|  | $b = 2^1$ | $b = 2^2$ | $b = 2^3$ | $b = 2^4$ | $b = 2^5$ | $b = 2^6$ | $b = 2^7$ |
|---|---|---|---|---|---|---|---|
| $r = 2^1$ | 61.0, 63.1 | 58.4, 60.4 | 57.6, 59.6 | 55.0, 56.8 | 53.3, 55.1 | 53.2, 54.8 | 54.8, 56.3 |
| $r = 2^2$ | 56.7, 58.7 | 55.2, 57.2 | 54.6, 56.5 | 52.3, 54.1 | 51.0, 52.7 | 51.1, 52.7 | 52.8, 54.3 |
| $r = 2^3$ | 55.4, 57.4 | 53.5, 55.5 | 53.2, 55.1 | 51.3, 53.0 | **50.1, 51.9** | 50.4, 52.0 | 52.2, 53.7 |
| $r = 2^4$ | 55.1, 57.2 | 53.2, 55.2 | 52.9, 54.9 | 51.2, 53.0 | 50.3, 52.1 | 50.6, 52.3 | 52.4, 54.0 |
| $r = 2^5$ | 56.0, 58.2 | 53.8, 56.0 | 53.7, 55.8 | 52.0, 53.8 | 50.9, 52.8 | 51.2, 52.9 | 52.9, 54.5 |
| $r = 2^6$ | 55.5, 57.7 | 53.9, 56.0 | 54.0, 56.0 | 52.2, 54.0 | 51.0, 52.9 | 51.3, 53.0 | 53.0, 54.6 |
| $r = 2^7$ | 56.3, 58.5 | 54.3, 56.5 | 54.1, 56.1 | 52.2, 54.1 | 51.1, 53.0 | 51.4, 53.1 | 53.2, 54.7 |

Table 3: Training and Development-set perplexities **without a document model**. Aggregation modes are: **radial concatenation** and **block summation**. Top-performing models are in **bold**.

|  | $b = 2^1$ | $b = 2^2$ | $b = 2^3$ | $b = 2^4$ | $b = 2^5$ | $b = 2^6$ | $b = 2^7$ |
|---|---|---|---|---|---|---|---|
| $r = 2^1$ | 61.3, 63.5 | 59.2, 61.3 | 58.4, 60.5 | 56.0, 58.0 | 54.3, 56.3 | 53.2, 55.2 | 52.6, 54.5 |
| $r = 2^2$ | 54.4, 56.4 | 53.4, 55.3 | 52.0, 53.9 | 49.6, 51.3 | 48.0, 49.8 | 47.2, 48.9 | 46.7, 48.4 |
| $r = 2^3$ | 50.5, 52.4 | 49.4, 51.2 | 47.9, 49.6 | 45.8, 47.3 | 44.4, 46.0 | 43.7, 45.3 | 43.4, 44.9 |
| $r = 2^4$ | 51.1, 53.0 | 50.2, 52.1 | 49.4, 51.2 | 47.5, 49.3 | 46.4, 48.2 | 45.8, 47.5 | 45.5, 47.2 |
| $r = 2^5$ | 51.1, 53.1 | 49.2, 51.1 | 48.4, 50.2 | 46.6, 48.3 | 45.6, 47.2 | 45.0, 46.6 | 44.7, 46.3 |
| $r = 2^6$ | 49.6, 51.5 | 48.2, 50.0 | 46.9, 48.5 | 45.2, 46.7 | 44.1, 45.6 | 43.5, 45.0 | **43.3, 44.7** |
| $r = 2^7$ | 49.5, 51.3 | 47.9, 49.7 | 46.8, 48.4 | 45.0, 46.4 | 44.0, 45.4 | 43.4, 44.9 | **43.3, 44.7** |

Table 4: Training and Development-set perplexities **without a document model**. Aggregation modes are: **radial concatenation** and **block concatenation**. Top-performing models are in **bold**.

|  | $b = 2^1$ | $b = 2^2$ | $b = 2^3$ | $b = 2^4$ | $b = 2^5$ | $b = 2^6$ | $b = 2^7$ |
|---|---|---|---|---|---|---|---|
| $r = 2^1$ | 60.5, 62.6 | 57.8, 59.8 | 56.9, 58.8 | 54.2, 56.0 | 52.5, 54.3 | 52.5, 54.1 | 54.1, 55.6 |
| $r = 2^2$ | 53.8, 55.7 | 52.2, 54.1 | 50.8, 52.6 | 48.2, 49.8 | 46.9, 48.5 | 47.1, 48.6 | 48.8, 50.1 |
| $r = 2^3$ | 50.0, 51.8 | 48.4, 50.2 | 46.9, 48.5 | 44.7, 46.2 | 43.6, 45.1 | 44.0, 45.4 | 45.6, 46.9 |
| $r = 2^4$ | 50.5, 52.4 | 49.1, 50.9 | 48.3, 50.1 | 46.5, 48.1 | 45.7, 47.3 | 46.3, 47.7 | 48.0, 49.4 |
| $r = 2^5$ | 50.5, 52.4 | 48.2, 50.0 | 47.3, 49.1 | 45.6, 47.1 | 44.8, 46.3 | 45.4, 46.8 | 47.1, 48.4 |
| $r = 2^6$ | 49.0, 50.9 | 47.3, 49.0 | 45.9, 47.4 | 44.2, 45.5 | **43.5, 44.8** | 44.0, 45.3 | 45.7, 46.9 |
| $r = 2^7$ | 49.0, 50.7 | 47.0, 48.7 | 45.9, 47.3 | 44.1, 45.4 | **43.5, 44.8** | 44.2, 45.4 | 46.0, 47.1 |

Table 5: Training and Development-set perplexities **with a document model**. Aggregation modes are: **radial concatenation** and **block summation**. Top-performing models are in **bold**.

|  | $b = 2^1$ | $b = 2^2$ | $b = 2^3$ | $b = 2^4$ | $b = 2^5$ | $b = 2^6$ | $b = 2^7$ |
|---|---|---|---|---|---|---|---|
| $r = 2^1$ | 61.3, 63.4 | 59.1, 61.3 | 58.4, 60.4 | 56.0, 58.0 | 54.3, 56.3 | 53.2, 55.2 | 52.5, 54.5 |
| $r = 2^2$ | 54.4, 56.3 | 53.3, 55.3 | 52.0, 53.9 | 49.5, 51.3 | 48.0, 49.8 | 47.1, 48.9 | 46.7, 48.4 |
| $r = 2^3$ | 50.5, 52.3 | 49.3, 51.1 | 47.9, 49.6 | 45.7, 47.3 | 44.4, 46.0 | 43.7, 45.3 | 43.3, 44.9 |
| $r = 2^4$ | 51.1, 53.0 | 50.1, 52.1 | 49.4, 51.2 | 47.5, 49.3 | 46.4, 48.2 | 45.8, 47.5 | 45.5, 47.2 |
| $r = 2^5$ | 51.1, 53.1 | 49.2, 51.1 | 48.4, 50.2 | 46.6, 48.3 | 45.6, 47.2 | 45.0, 46.6 | 44.7, 46.3 |
| $r = 2^6$ | 49.6, 51.5 | 48.2, 50.0 | 46.9, 48.5 | 45.1, 46.7 | 44.1, 45.6 | 43.5, 45.0 | 43.3, 44.7 |
| $r = 2^7$ | 49.5, 51.3 | 47.9, 49.7 | 46.8, 48.4 | 45.0, 46.4 | 44.0, 45.4 | 43.4, 44.9 | **43.2, 44.7** |

# E    ABBREVIATED TRAINING LOGS

| Step | Perplexity | Samples |
|---|---|---|
| Init-0 | 43.33 | `<sod>, not was.<eod><sod>Did have<eod><sod>" isI stop<eod><sod> don't to that you only<eod>` |
| Tune-1 | 42.87 | `<sod>, have!.<eod><sod>B this<eod><sod>The and when everything<eod><sod> go to of the C<eod>` |
| Tune-2 | 42.67 | `<sod>, have!.<eod><sod>Get not<eod><sod>The and if he's<eod><sod> her to of theI<eod>` |
| Tune-3 | 42.58 | `<sod>, have this the<eod><sod> may for<eod><sod> .<eod><sod> here doing<eod><sod> out a to` |
| ... | ... | ... |
| Tune-9 | 42.46 | `<sod>, have this the<eod><sod> away!<eod><sod> .. itum<eod><sod>! a I of which<eod>` |
| Tune-10 | 42.44 | `<sod>, have this the<eod><sod> lastI.<eod><sod> in get four<eod><sod>! I that a away<eod>` |
| Train-11 | 37.74 | `<sod>I of do,<eod><sod> after is.<eod><sod>"[ long<eod><sod> then you want a tell<eod>` |
| Train-12 | 36.18 | `<sod>I will just the<eod><sod>Here on, is in now believe<eod><sod> an I think toer.<eod>` |
| ... | ... | ... |
| Train-158 | 23.74 | `<sod>I had the next time ever got to say that were,,<eod><sod> been, for a ho population and` |
| Train-159 | 23.73 | `<sod>I had the only one don't, and all in time way of each other.<eod><sod>more than 16!<eod>` |
| Train-160 | 23.73 | `<sod>I can't remember a lot problem the same way.<eod>` |
| Train-161 | 23.72 | `<sod>I had the next time, to a number of was seven in fact and his<eod>` |
| Train-162 | 23.72 | `<sod>I had the next time be no one would ever much believe<eod>` |
| Train-163 | 23.71 | `<sod>I can't see it. ''<eod><sod>The end of my problem, sea is him and she say long P W days` |
| Train-164 | 23.70 | `<sod>I can't see it. ''<eod><sod>The end of her too, where the same as Advice was published` |
| Train-165 | 23.70 | `<sod>I had the next timet an hour<eod><sod>is about county, where are you to cry in 2005` |
| Train-166 | 23.68 | `<sod>I had the door, Sam we are two talking from life of men and one time away<eod>` |
| Train-167 | 23.69 | `<sod>I had the only one four who is it to think means of The two and a former in some home` |

Table 6: Abbreviated logs from training this work's 'best' model on the 10M data set ($r = 2^3$), demonstrating how it is often helpful to view samples of model output alongside performance metrics as a means of assessing the quality of a model's optimization. This becomes especially important as different data sets and vocabularies are explored, as the lowest numerical values of perplexity don't necessarily correspond to the most cogent text.

