# OpenReview forum: "Explicit Foundation Model Optimization with Self-Attentive Feed-Forward Neural Units"
_ICLR.cc/2024/Conference — ICLR 2024 Conference Withdrawn Submission_

### Official Review · Reviewer_QpKp · 2023-10-17

**Soundness:** 3 good
**Presentation:** 2 fair
**Contribution:** 3 good
**Rating:** 3
**Confidence:** 3

**Summary:**

The authors introduce self-attentive feed-forward unit (SAFFU) layers, a mathematically simplified version of regular self-attention transformer layers. The redefined formalism allows them to find (approximate) explicit solutions for the weight matrices in the neural network. Using this solution the SAFFU networks may be initialized better, outperforming solutions found by backpropagation from the get-go. Fine-tuning the better initialization with backpropagation (called warm-start) yields even better results. The evaluation has been performed on the BabyLM NLP dataset and a pretraining problem with a deep SAFFU network. The authors report substantially improved perplexity results.

**Strengths:**

- Clever mathematical framework for avoiding a large number of backpropagation iterations
- Possibly significantly improved initialization of transformer layers; hard to judge based on the experimental evaluation

**Weaknesses:**

- We ask: “how can these challenges can be overcome to ensure models are reliable, interpretable, and efficient?” - I am not convinced that the papers provides answers to the first two sub-questions as the authors do not analyze the robustness or interpret its partial results
- The sample texts provided in the appendix are not really convincing with respect to generation performance. Most samples do not have even a correct placement of <sod/eod> tokens, the actual texts are gibberish
- The experimental evaluation of the approach should be significantly expanded with respect to the following issues:
    * Clarification of what the backpropagation case (Fig. 1) constitutes. Is this pure backpropagation on SAFFU units or backpropagation on default transformer layers? If the former, please include a comparison against default transformers, if the latter please include a backpropagation-only trained SAFFU experiment
    * Different datasets to demonstrate the general applicability of this approach. A vision transformer may be a good starting point.
    * Figure 1, right has only one y-axis value making it impossible to judge the scale of the values.
- Structurally, the bit-cipher algorithm is a fundamental part of the work and not merely a side note in the appendix. It seems meaningful to include the algorithm in the body of the manuscript.
- While mathematically rigorous it stands to reason that the paper would benefit from a more intuitive explanation of Sections 2 and 3
- The clarity may be improved by better writing and wording

Minor issues:

- It may be advisable to split the current introduction into two separate subsections for introduction and related work

**Questions:**

- The paper claims efficiency improvements, but does not evaluate any computational characteristics. Is there a significant overhead in using SAFFU in terms of wall-time, memory or FLOPs?

---

### Official Review · Reviewer_kr31 · 2023-10-30

**Soundness:** 3 good
**Presentation:** 1 poor
**Contribution:** 1 poor
**Rating:** 1
**Confidence:** 3

**Summary:**

The work develops a combination of attention and feedforward layer which can be initialized with data to then train faster once backpropagation is used on the remaining data.

**Strengths:**

It is an interesting concept to optimize the self-attentive feedforward units before actual training with backpropagation.

**Weaknesses:**

- No baselines for regular transformer backpropagation training. As such, it is not clear how effective the new architecture is.
- Ablations done on 1M tokens. It is not clear if this is sufficient to ablate components.
- Experiments were done with at most 100M tokens. As such, scaling behavior is not clear.

**Questions:**

How does the empirical performance of your method compare to transformers on the same data?

---

### Official Review · Reviewer_Nd49 · 2023-10-30

**Soundness:** 1 poor
**Presentation:** 1 poor
**Contribution:** 2 fair
**Rating:** 3
**Confidence:** 4

**Summary:**

The submission presents an architecture based on a variant of self-attention, and a way of initializing the weights that layer. That pre-initialization has a closed-form formula, computed from the co-occurrence matrix of tokens.
Experiments on a 10-million token dataset from the BabyLM Challenge, show this initialization performs better than random initialization.
Additional experiments are performed to investigate the role of some hyperparameters, and scaling up to a 100M token dataset.

**Strengths:**

An architecture for sequence-to-sequence learning that works well on smaller corpora, and can be trained using fewer computation resources, could be impactful. A better-than-random pre-initialization, speeding up training, is also interesting in that context.

**Weaknesses:**

Originality
--------------

1. There should be citations about self-attention mechanisms

Quality
----------
The major issue is the lack of comparisons or baselines:
1. no comparison with other, existing models is performed at all: including n-gram based, RNNs, transformers...
2. no comparison on the efficiency, in terms of data or compute used

So the published perplexity numbers are meaningless, and it is impossible to assess the usefulness of the proposed method.

Clarity
---------
The paper was challenging to read through, and many points remain unclear. In particular:
1. The dimensions of the various tensors is confusing, in particular what `K` and `M` represent.
2. Section 2.1 implies that it's just reparametrizing and rearranging terms of a regular self-attention layer, but it's really a different model, although it shares similarities:
  - W_q W_k^T could be of a lower rank than their combined product
  - Going from X W X^T to X X^T W is again a different model, maybe closer to attention between X and W^T X, with W_k = W_q = Id, not self-attention
  - Adding the log around the softmax before multiplying with the "value" is also a different model
  - Using softmax (a second time) as a replacement for the activation function, when it's not elementwise unlike what is generally recognized as an activation function, again makes it a model that may not be expected to behave like a "self-attention" mechanism
  - Despite all of that, it is labeled "self-attention" in Fig. 1, for instance.
3. The abstract states this method is an "alternative for optimizing neural networks", but the pre-initialization is not a complete alternative for backprop, and is only applicable for layers of a particular form. In particular, the "feed-forward" blocks depicted in Figure 1 are not described as using this pre-initialization. This is quite misleading.
4. The title mentions "foundation models", but the datasets used (and, presumably, the models) are not at a scale large enough to be comparable with foundation models.
5. The role of `h` as the layer's "head" is also unclear, as is the reason it was introduced, and how it affects the behaviour in practice.

Significance
----------------
In the absence of baselines, it's not possible to assess the significance.

Minor points
-----------------
1. the "red-blue" result is not introduced or defined
2. double-check the bibliography for duplicate or ill-formatted references

**Questions:**

1. How does that method and architecture perform against baselines?
2. What is the computational cost of computing the pre-initialization of weights?
3. Is there a sign issue between the theorem text and the equations?

**Details Of Ethics Concerns:**

Large parts of the submission (theorem, full paragraphs) are shared with another submission I'm reviewing ([7828](https://openreview.net/forum?id=OFgOmMlVUY)), and neither acknowledge the other one.

There seems to be vague, maybe left-over references to "discovering the priming number" and their "work now pics up from that point" at the beginning of section 2. This may hint about this paper being a follow-up on the other one, but there is still significant overlap.

The other paper also has a different set of experiments in a different framing, so it does not look like it was a preliminary version that is superseded by this one.

---

### Official Review · Reviewer_VevR · 2023-11-02

**Soundness:** 2 fair
**Presentation:** 1 poor
**Contribution:** 3 good
**Rating:** 5
**Confidence:** 3

**Summary:**

This paper proposes self-attentive feed-forward unit layers, SAFFU, for building
hyper-efficient transformers with efficient training by leveraging explicit solution.

Experimental results show SAFFU-transformer can achieve high performance with less data when
applying explicit solutions.

**Strengths:**

The motivation to propose explicit optimization techniques for reducing LLM training cost is reasonable
and SAFFU layers for building efficient transformer is interesting.

SAFFU layers are formulated with proof and its initialization is well described.
A SAFFU-based transformer architecture is proposed for language task.

Ablation on a small set of data samples is conducted to show the advantages of SAFFU-based transformer.

**Weaknesses:**

The claims in this paper seem not well supported by the model and experiments. e.g., 1, the paper claims explicit foundation model optimization with SAFFU. But SAFFU-Transformer only has 10M parameters. The experiments in the paper are not enough to support SAFFU-Transformer is a foundation model that can be generalized to many different language modeling tasks. 2, back-propagation only leads to generalized model with the explicit solution. Simple ablations on SAFFU-Transformer may not hold for LLMs.

The presentation of this paper can be improved. I need to keep checking appendix to understand the paper since some formula in the main paper are defined in the appendix. For example, there is no explanation of Eq. 7 in the main paper.

Hard to find the support for some arguments. For A_m, there is no answer why the order of self-attention can be exchanged.

There are no baselines compared in the paper. I can see good initialization methods [1][2] can be the straightforward baselines.

Even with explicit solution, back-propagation is still needed in the paper to train SAFFU. Then why the proposed method is better than finding good initialization points? And how long need to train SAFFU-Transformer after applying explicit solution?

[1] On Warm-Starting Neural Network Training
[2] A weight initialization method for improving training speed in feedforward neural network

**Questions:**

Please see weaknesses. I will change my rating based on the response.